# Laboratory Diagnostic of Acute Kidney Injury and Its Progression: Risk of Underdiagnosis in Female and Elderly Patients

**DOI:** 10.3390/jcm12031092

**Published:** 2023-01-30

**Authors:** Thea Sophie Kister, Maria Schmidt, Lara Heuft, Martin Federbusch, Michael Haase, Thorsten Kaiser

**Affiliations:** 1Institute of Laboratory Medicine, Clinical Chemistry and Molecular Diagnostics (ILM), University of Leipzig Medical Center, 04103 Leipzig, Germany; 2Institute for Human Genetics, University of Leipzig Medical Center, 04103 Leipzig, Germany; 3Medical Faculty, Otto-von-Guericke University Magdeburg, 39106 Magdeburg, Germany; 4Department of Kidney Diseases and Hypertension, Hannover Medical School, 30625 Hannover, Germany; 5DIAMEDIKUM Kidney Care Center Potsdam, 14473 Potsdam, Germany; 6Institute for Laboratory Medicine, Microbiology and Pathobiochemistry, University Hospital Ostwestfalen-Lippe (UK-OWL) of Bielefeld University, Campus Klinikum Lippe, 32756 Detmold, Germany

**Keywords:** acute kidney injury, CDSS, age and sex, laboratory diagnostics, KDIGO, serum creatinine, women, elderly

## Abstract

Acute kidney injury (AKI) is a common disease, with high morbidity and mortality rates. In this study, we investigated the potential influence of sex and age on laboratory diagnostics and outcomes. It is known that serum creatinine (SCr) has limitations as a laboratory diagnostic parameter for AKI due to its dependence on muscle mass, which may lead to an incorrect or delayed diagnosis for certain patient groups, such as women and the elderly. Overall, 7592 cases with AKI, hospitalized at the University of Leipzig Medical Center (ULMC) between 1st January 2017 and 31st December 2019, were retrospectively analyzed. The diagnosis and staging of AKI were performed according to the Kidney Disease: Improving Global Outcomes (KDIGO) guidelines, based on the level and dynamics of SCr. The impact of sex and age was analyzed by the recalculation of a female to male and an old to young SCr using the CKD-EPI equation. In our study cohort progressive AKI occurred in 19.2% of all cases (*n* = 1458). Female cases with AKI were underrepresented (40.4%), with a significantly lower first (−3.5 mL/min) and last eGFR (−2.7 mL/min) (*p* < 0.001). The highest incidence proportion of AKI was found in the [61–81) age group in female (49.5%) and male (52.7%) cases. Females with progressive AKI were underrepresented (*p* = 0.04). By defining and staging AKI on the basis of relative and absolute changes in the SCr level, it is more difficult for patients with low muscle mass and, thus, a lower baseline SCr to be diagnosed by an absolute SCr increase. AKIN1 and AKIN3 can be diagnosed by a relative or absolute change in SCr. In females, both stages were less frequently detected by an absolute criterion alone (AKIN1 ♀ 20.2%, ♂ 29.5%, *p* < 0.001; AKIN3 ♀ 13.4%, ♂ 15.2%, *p* < 0.001). A recalculated SCr for females (as males) and males (as young males) displayed the expected increase in AKI occurrence and severity with age and, in general, in females. Our study illustrates how SCr, as the sole parameter for the diagnosis and staging of AKI, bears the risk of underdiagnosis of patient groups with low muscle mass, such as women and the elderly. A sex- and age-adapted approach might offer advantages.

## 1. Introduction

Acute kidney injury (AKI) is a life-threatening interdisciplinary disease that remains partly undetected and undertreated. With an incidence proportion ranging from 7.2% to 31.3%, AKI is a common complication and requires timely and professional management [1,2]. Associations with increased short- and long-term mortality rates, longer lengths of hospitalization, and higher rates of renal replacement therapy have been observed in several studies [1,3,4,5]. In particular, the progression of AKI to a higher stage according to the Kidney Disease: Improving Global Outcomes (KDIGO) guidelines is associated with significantly worse outcomes, such as higher rates of mortality and renal replacement therapy [3]. In recent years, interest in the implementation of electronic information and alert systems to support decision making by physicians has generally increased [6]. A clinical decision support system (CDSS) can support the recognition, monitoring, and timely treatment of AKI and provide an opportunity to improve management and, thus, clinical outcomes [7,8]. However, so far, the use of CDSS has shown a minor impact on the outcome of AKI [2,9]. Currently, AKI is defined and staged according to the KDIGO guidelines based on increased serum creatinine (SCr) and/or reduced urine volume [10]. However, this guideline, which is mainly based on the laboratory parameter SCr in daily practice, does not stratify detection by sex or age. It is well known that the concentration of SCr is strongly affected by muscle mass and is, therefore, not a universal parameter of renal function [11,12]. Patients with a physiologically lower muscle mass, which is, on average, more frequently encountered in females and the elderly, might benefit from an age- and sex-specific diagnostic process. It is generally accepted that, for this reason, renal function should not be assessed on the basis of SCr but on the basis of the glomerular filtration rate, the calculation of which includes age and sex [11].

In this study, we systematically investigated the potential influence of age and sex on the laboratory diagnostic of AKI and its progression by retrospectively comparing the detection and progression of AKI between both sexes stratified by age groups. In addition, we calculated a hypothetical SCr based on the estimated glomerular filtration rate (eGFR), which adjusts the original SCr by age and sex and, thus, compares the potentially less age- and sex-biased detection process with the original one. Options to achieve appropriate management through an adequate diagnostic procedure that takes into account age and sex, in addition to existing laboratory parameters, were explored.

## 2. Materials and Methods

For our retrospective cohort study, 63,239 medical cases at the University of Leipzig Medical Center (ULMC) (1451 beds), hospitalized between 1 January 2017 and 31 December 2019, met the inclusion criteria and were further analyzed. A case was defined as a continuous hospital stay of one patient. Inclusion criteria consisted of a patient age ≥ 18 years (*n* = 129,232), ≥ two creatinine measurements (*n* = 64,833), and the absence of the code N18.5 (dialysis-dependent chronic kidney disease) according to the International Statistical Classification of Diseases and Related Health Problems—German Modification (ICD-10-GM) [13] (*n* = 63,239). According to the KDIGO guidelines using only SCr levels, 7592 cases with potential AKI were detected and staged using the acute kidney injury network criteria (Table 1, Figure 1). AKI progression was present if the first AKI stage during hospitalization was lower than the maximum AKI stage per case. AKI was not considered if dialysis was performed up to 72 h before SCr measurement.

Acquisition of data and statistics: SCr measurements were performed in serum on a Cobas 8000 Analyzer module c 701 (Roche, Mannheim, Germany; Creatinine Plus Ver. 2kit, enzymatic method), available from the laboratory information system LabCentre (i-SOLUTIONS Health, Version 2022.01 HF 1) at ULMC. Additional clinical data (e.g., ICD-10-GM codes, admission and discharge timestamps, dialysis procedures, mortality) for the included cases were retrieved from the Clinical Information System (SAP Software Solutions, Walldorf, Germany [14]).

A case-based analysis was performed using Microsoft Excel for Office 365 ProPlus (Microsoft Corporation, Redmond, WA, USA); further statistical analysis was performed using R 4.0.2 [15], with the addition of the reshape2 package [16]. A Mann–Whitney U test was used to compare group medians of continuous variables (e.g., between progressive and non-progressive AKI cases and between males and females). A chi-square test was used to evaluate categorical variables accordingly. Differences at an alpha level of 0.05 were deemed significant. The eGFR was determined using the Chronic Kidney Disease Epidemiology Collaboration equation (CKD-EPI) [17] for our predominantly Caucasian cohort. Staging of AKI holds that the intensity of SCr increases over a given timeframe and can be absolute or relative. AKIN1 and AKIN3 can be reached by relative and absolute criteria, whereas AKIN2 requires only a relative increase. The mode of AKI staging (relative, absolute, both) for each AKI instance was flagged, and the proportion of each mode per stage was compared between males and females.

Hypothetical recalculation of SCr: The impact of sex on the detection of AKI was analyzed as follows using all data points of female cases (165,267 measurements): eGFR values (determined on the basis of original creatinine, female sex, and age using the CKD-EPI equation [17]) were back-calculated into creatinine using the male factor for sex, thus, yielding a “male” creatinine value for a person of the same age and eGFR as the original (e.g., female, age 40 years, SCr 100 μmol/L -> eGFR 60.7 mL/min/1.73m2 -> male, 40 years, SCr 159.7 μmol/L). This corresponds to a hypothetical value as if the patient with a certain eGFR were male instead of female (female-as-male cohort) (Appendix A). The impact of age on the detection of AKI was analyzed using all data points of male cases (231,365 measurements) and the same approach as above: eGFR values (determined based on original creatinine, male sex, and age) were back-calculated into creatinine using a fixed age of 30 years, thus, yielding a “young” creatinine for a male person with the same eGFR as the original (males-as-young cohort). Both cohorts with recalculated creatinine were used for AKI detection and staging as with the original creatinine input.

## 3. Results

### 3.1. Study Cohort Characteristics of Females (♀) and Males (♂) Regarding Non-Progressive and Progressive AKI

From our cohort of 63,239 investigated inpatient cases, AKI was diagnosed in 7592 cases (12.0%), which defined our final study cohort (Figure 1). For all the AKI cases, sex-specific differences were present: Males with AKI were on average 4.3 years younger than females (*p* < 0.001), had 1.4 days longer length of hospitalization (*p* = 0.04), and a higher first (+3.5 mL/min) and last eGFR (+2.7 mL/min) (*p* < 0.001). The time to first AKI during hospitalization (♀, 4.7 days; ♂, 4.5 days; *p* = 0.07) and the mortality rate (♀, 22%; ♂, 23%; *p* = 0.31) did not differ significantly. Considering only the progressive AKI cases, males were on average 4.4 years younger than females (*p* < 0.001) and had a shorter time to first AKI during hospitalization (♀, 4.9 days; ♂, 4.5 days; *p* = 0.04). The first eGFR (+3.9 mL/min, *p* = 0.18), the last eGFR (-1.1 mL/min, *p* = 0.50), the length of hospitalization (♀, 24.7 days; ♂, 26.3 days; *p* = 0.49), and mortality rate (♀, 45.5%; ♂, 45.0%; *p* = 0.91) did not differ significantly (Table 2). Progressive AKI (AKIN1 to 2, AKIN1 to 3, and AKIN2 to 3) was found in 1458 cases (19.2% of AKI cases), with the incidence proportion significantly higher in male cases (20.0%) than in female cases (18.1%) (*p* = 0.04) (Table 2, Appendix A). Comparing progressive AKI with non-progressive AKI, there were no sex-related differences. The outcome of progressive AKI cases was significantly worse in both sexes. For example, female cases with progressive AKI were on average younger at first AKI detection during hospitalization and showed a higher mortality rate and longer hospital stays than cases without progression (*p* < 0.001). The last eGFR was significantly lower in cases with progressive AKI (*p* < 0.001), whereas the first eGFR was not significantly different. In addition, the time to the first AKI was comparable between progressive and non-progressive cases and showed no significant differences. The observation of associated comorbidities showed a significantly higher proportion of myocardial infarction, cardiac insufficiency, sepsis, shock, and liver cirrhosis in progressive AKI cases than in non-progressive AKI cases. Similar results were obtained for male cases (Appendix A).

### 3.2. Adding Age to AKI Detection

Cases were divided into four different age groups ([18–41), [41–61), [61–81), and [81–max]). The incidence proportion of AKI increased with age, with the [61–81) age group contributing the most cases of AKI to the study cohort in male (52.7%) and female (49.5%) cases. The same age group contained the most progressive AKI cases in both sexes. The incidence proportion of AKI was significantly lower in females than in males for the [18–41) (♀, 4.7%; ♂, 5.8%; *p* = 0.02), [41–61) (♀, 9.0%; ♂, 11.2%; *p* < 0.001) and [81–max] (♀, 14.3%; ♂, 17.4%; *p* < 0.001) age groups (Table 3). Progressive AKI cases per age group did not show the same trend as the incidence proportion, but there was a trend for lower proportions in females in each age group (Table 4). With regard to the definition of the three AKI stages according to the KDIGO guidelines, AKIN1 and AKIN3 were reached through a relative or absolute increase in SCr (Table 1). Female cases reached AKIN1 (♀, 20.2%; ♂, 29.5%; *p* < 0.001) and AKIN 3 (♀, 13.4%; ♂, 15.2%; *p* < 0.001) at a smaller proportion through the absolute staging criterion alone than did male cases (Table 5). The number of undetected AKI cases according to KDIGO criteria remained unknown; however, consequently, an assessment of whether the relative criterion thoroughly detects females was not possible without a detailed inspection of all health records. Considering the distribution of AKI stages within the age groups, a shift of female-as-male cases toward male cases could be observed in cases with AKIN3 (Figure 2, Appendix A).

According to the KDIGO guideline, three criteria for the detection of AKI are possible: (1) An increase in the last seven days by at least 1.5 times baseline. (2) Absolute increase in the last two days. (3) Creatinine limits exceeded (if also relative/absolute increase according to (1) or (2)). Especially, criterion 2 should be more difficult to achieve in people with less muscle mass. Hypothetically, women and older people on average would have proportionately less grouping over criterion 2. AKIN1 and 3 are staged through criteria 1 or 2. AKIN2 is detected exclusively via criterion 1 and, thus, remains irrelevant for sex differences.

### 3.3. Impact of SCr Recalculation

We were able to compare the original laboratory diagnosis of AKI with a modified one by calculating a hypothetical SCr from the originally obtained eGFR (via CKD-EPI [17]) and staging AKI, as previously stated. Figure 2 gives the proportion of initial and maximum AKIN3 in these recalculated cohorts and the original ones for females and males (see Appendix A for the proportions of all AKI stages). The percentage of initial and maximum AKIN3 in the original cohorts generally decreased with age. The same held true for the recalculated female-as-male cohort. Recalculated males-as-young demonstrated an increase in these percentages with age. With the exception of the [18–41) age group, the female-as-male cohort shifted toward a pattern observed for males (Figure 2).

## 4. Discussion

AKI is a common disease associated with high mortality and morbidity rates [1,4,5,18,19,20]. In particular, progressive AKI, to date a less studied phenomenon, has a strong prognostic value and negative clinical course. The general importance of the timely detection and adequate treatment of progression has already been described in a previous study [3]. To the best of our knowledge, this study is the first to analyze sex and age differences in the laboratory-based diagnosis of progressive AKI using a recalculation based on the eGFR to convert SCr for opposite sex and age groups, clearly illustrating the risk of underdiagnosis in female and elderly patients.

It is well known that the SCr is influenced by several parameters besides kidney function (muscle mass, diet, physical activity, and medications) and decreases constantly from the third to the ninth decade of life [11,21,22]. Several equations for the calculation of an eGFR have been proposed that consider SCr and various additional parameters [23]. The CKD-EPI equation, developed in 2009 and applied in this study, uses age, sex, and race in addition to SCr [17] to compensate for the limitations of SCr alone. We saw an overall smaller incidence proportion of AKI and its progression in females and a tendency for a decrease in these endpoints with increased age. This could be explained by biological and lifestyle factors [24] but also by sex bias in laboratory AKI diagnostics. Our study also investigated sex and age bias in AKI detection and its progression by using a calculation of an adjusted SCr. We were thereby able to demonstrate a shift in AKI detection in females toward a laboratory-based diagnosis more similar to men. Additional circumstantial evidence comes from the different ways in which AKI stages are reached (by a relative or absolute SCr increase or both) in females and males. This raises the question of whether more female cases are at risk for AKI and its progression than are currently detected by following the KDIGO guidelines.

It seems reasonable that women and older patients might be disadvantaged under the KDIGO guidelines due to lower muscle mass on average and, therefore, lower baseline SCr levels and the potential for SCr increase. With a surplus of women in the elderly population, both in Saxony and globally [24,25,26], as well as a general increase in lifespan, these disadvantages are becoming more relevant. Muscle mass decreases with age, so-called senile muscular atrophy [27], which is a consequence of numerous aging processes, including hormonal changes, reduced physical activity, oxidative stress, and malnutrition. The sex hormone testosterone, in particular, has an essential impact on the development of sarcopenia and, hence, the progressive loss of function of muscle mass or muscle mass wasting in the elderly [28]. As Yoo et al. also note, testosterone reduction with aging could be a reason and a risk factor for the overestimation of kidney function in older men [12]. This could explain the decrease in differences in endpoints between males and females with increasing age.

Cobo et al. discussed sex differences, which may have an impact on the epidemiology and progression of renal dysfunction. Estrogen appears to act nephroprotectively through several mechanisms [29]. Obviously, this must also be considered with regard to our results. Despite this, by the recalculation of female creatinine to female-as-male creatinine, we were able to show that not every woman is detected with AKI according to KDIGO guidelines and that physiological differences are not solely responsible for the sex imbalance in AKI patients. The diagnosis based on unadjusted SCr alone accounts, in part, for the differences.

Recommendations for improved AKI diagnostics include the use of calculated reference change values of SCr, adjusted to the respective baseline creatinine. This could increase the sensitivity of laboratory AKI diagnostics, both for patients with high and low baseline creatinine levels [30]. The inclusion of the biomarker cystatin C also seems useful because it represents renal function in real time, independent of muscle mass or diet [31]. However, clinical standardization and evaluation in the context of AKI still need to be established. A known complication of AKI is an increased risk of developing CKD [32,33]. The underdiagnosis of AKI episodes in women might lead to an increased incidence of CKD. This hypothesis is supported by several studies [34,35]. A systematic review by Carrero et al. revealed a higher incidence proportion of CKD in women than in men. James et al. showed an independent association of, among other factors, age and female sex with CKD [36]. A higher risk of progression from CKD to end-stage renal disease (ESRD) for men (i.e., fewer women receiving renal replacement therapy (RRT)) might, in part, be attributed to societal factors [35,37].

These aspects should be considered when defining and staging AKI according to the KDIGO guidelines, which mainly use SCr as a single laboratory parameter. Currently, sex and age are not taken into account by the KDIGO guidelines when interpreting SCr kinetics [10], which results in a disadvantage for women and the elderly. We recommend the amendment of the guidelines to incorporate age and sex for the interpretation of creatinine kinetics based on thorough systematic evidence of the matter. Further additions might include eGFR-based diagnostics or markers, as mentioned above. The limitations of this study are its exclusively retrospective analysis and the non-inclusion of urine excretion as an AKI criterion. Regarding urine excretion, the KDIGO guidelines already mention that the excretion criteria for AKI diagnosis and staging have been less well validated, and this parameter is usually not captured in administrative databases and often not even measured [10]. Thus, we judge SCr to be the current better alternative for addressing our study question. Furthermore, we only considered inpatient cases, and long-term outcomes were not analyzed. Patients initially hospitalized for AKI were unlikely to be detected by our AKI algorithm and were, therefore, not part of our cohort.

To date, AKI seems to have been an underrated clinical condition [3]. Its definition and staging are complex, and using a CDSS for the detection of AKI would provide automatic detection and staging, as well as recommendations for management and timely intervention [3,6].

Our study supports the hypothesis of the underdiagnosis of AKI in females and the elderly. To the best of our knowledge, our study is the first to recalculate SCr to the opposite sex and lower age vie eGFR, providing a feasible recalculation tool. The method we used could be a first step toward the consideration of sex- and age-specific SCr increases in AKI detection and staging, pending evaluation in the clinical routine. We can show that the exclusive consideration of SCr as a parameter and the use of static absolute criteria to define an AKI may result in a biased diagnosis. These findings should raise awareness not to uncritically apply diagnostic criteria to all patients regardless of sex, age, as well as other criteria such as muscle mass in a clinical context. This could help to ensure that patients with an increased risk of kidney injury underestimation receive a more equitable diagnosis through additional laboratory parameters (e.g., CysC) and provide adequate therapy. We want to emphasize that our study does not propose a new AKI definition, which is currently clinically applicable. Rather, we wish to reiterate that the current criteria for AKI diagnosis may lead to possible underdiagnosis. We believe that the criteria for AKI diagnosis should be expanded to include a factor that compensates for sex or that a measure of filtration that is unlikely to be influenced by sex should be used. Further studies in this regard are urgently needed.

## Figures and Tables

**Figure 1 jcm-12-01092-f001:**
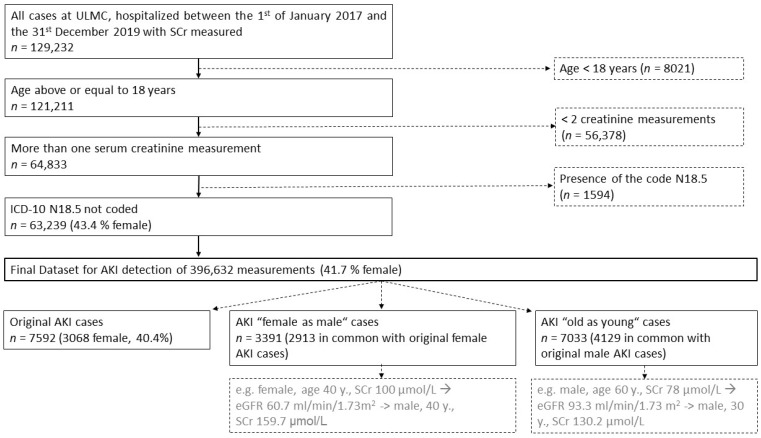
Study cohort with inclusion and exclusion criteria. *n*—number of cases.

**Figure 2 jcm-12-01092-f002:**
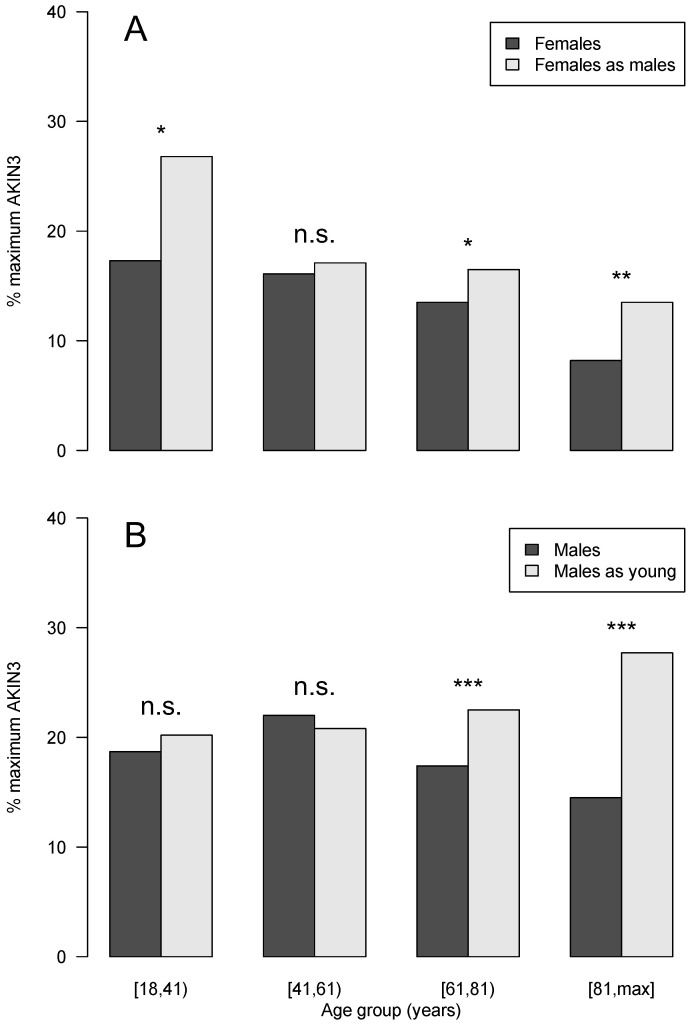
Proportion of AKI cases reaching AKIN3 during hospitalization per age group. Panel (**A**) compares females to the recalculation cohort females-as-males. Proportions decline with age group but are consistently higher in the recalculated cohort. Panel (**B**) compares males to the recalculation cohort males-as-young. Differences in proportions increase with age (*p*-value n.s. ≥ 0.05, * <0.05, ** <0.01, and *** <0.001).

**Table 1 jcm-12-01092-t001:** AKI staging according to the KDIGO guidelines [10].

Stage	Serum Creatinine	Urine Output
AKIN1	1.5–1.9 times baseline, OR ≥0.3 mg/dL (≥26.5 μmol/L) increase	<0.5 mL/kg/h for 6–12 h
AKIN2	2.0–2.9 times baseline	<0.5 mL/kg/h for ≥12 h
AKIN3	3.0 times baseline, OR increase in serum creatinine to ≥4.0 mg/dL (≥353.6 μmol/L), OR initiation of renal replacement therapy, OR in patients <18 years, decrease in eGFR to <35 mL/min per 1.73 m^2^	<0.3 mL/kg/h for ≥24 h, OR anuria for ≥12 h

**Table 2 jcm-12-01092-t002:** Comparison of female and male patient characteristics considering all AKI cases and progressive AKI cases. Variables are given as medians [interquartile range] or percentages. *n*—number of cases. Significant *p*-values (< 0.05) highlighted as bold.

	All AKI Cases, *n* = 7592			Progressive AKI Cases, *n* = 1458		
	Female	Male	*p*-Value	Female Progression	Male Progression	*p*-Value
Incidence proportion, *n*, %	3068, 40.4	4524, 59.6		554, 18.1	904, 20.0	**0.039**
**Basic patient characteristics**						
Age (years)	72.0 [60.4–80.9]	67.7 [57.9–77.8]	**<0.001**	70.7 [59.0–79.5]	66.0 [57.0–75.9]	**<0.001**
Total length of hospitalization (days)	16.2 [9.1–29.6]	17.6 [9.1–31.6]	**0.038**	24.7 [14.2–43.1]	26.3 [14.9–42.6]	0.491
First eGFR (mL/min/1.73 m²)	59.9 [36–85.4]	63.4 [39.3–88.2]	**<0.001**	58.0 [36.1–85.1]	61.9 [37.7–88.1]	0.177
Last eGFR (mL/min/1.73 m²)	53.2 [32.7–82.2]	55.9 [35.7–84.5]	**<0.001**	44.9 [25.2–74.4]	46.0 [25.7–75.7]	0.502
Time to first AKI during hospitalization (days)	4.7 [1.9–10.8]	4.5 [1.8–9.9]	0.073	4.9 [1.9–11.9]	4.5 [1.8–9.4]	**0.037**
In-hospital mortality	22.0	23.0	0.310	45.5	45.0	0.905
**Comorbidities**						
I10.—Hypertension	48.2	46.4	0.129	47.7	46.9	0.822
E11.—Diabetes mellitus	31.8	34.4	**0.022**	31.4	35.2	0.156
E86.—Exsiccosis	4.8	4.4	0.466	6.0	3.4	**0.031**
R57.—Shock	17.2	22.3	**<0.001**	37.9	44.3	**0.020**
I25.—Coronary heart disease	13.5	23.4	**<0.001**	11.2	23.5	**<0.001**
I21.—Myocardial infarction	3.2	4.9	**0.001**	3.4	6.2	**0.028**
I50.—Cardiac insufficiency	28.9	28.0	0.435	33.9	34.6	0.832
A41.—Sepsis	19.9	24.9	**<0.001**	38.8	41.9	0.263
K74.—Liver cirrhosis	5.1	5.1	0.993	8.7	9.0	0.865

**Table 3 jcm-12-01092-t003:** Comparison of AKI in different age groups with regard to sex. A total of 63,239 cases were considered for the detection of AKI. *n*—number of cases.

Age Group	All AKI (*n* = 7592, 12.0%)	Female (*n* = 3068, 11.2%)	Male (*n* = 4524, 12.6%)
[18–41) *	470, 5.2%	202, 4.7%	268, 5.8%
[41–61) ***	1765, 10.4%	596, 9.0%	1169, 11.2%
[61–81) n.s.	3902, 14.0%	1519, 13.5%	2383, 14.3%
[81–max] ***	1455, 15.6%	751, 14.3%	704, 17.4%

Asterisks indicate significant differences between female and male AKI cases (*p*-value n.s. ≥ 0.05, * <0.05, and *** <0.001).

**Table 4 jcm-12-01092-t004:** Comparison of the proportions of progressive AKI cases within different age groups, stratified by sex. A total of 7592 AKI cases were considered for detection of progression. *n*—number of cases.

Age Group	Total Progressive AKI (*n* = 1458, 19.2%) *	Female (*n* = 554, 7.3%)	Male (*n* = 904, 11.9%)
[18–41) n.s	85, 18.1%	36, 17.8%	49, 18.3%
[41–61) n.s	389, 22.0%	124, 20.8%	265, 22.7%
[61–81) n.s	748, 19.2%	281, 18.5%	467, 19.6%
[81–max] n.s	236, 16.2%	113, 15.0%	123, 17.5%

Asterisks indicate significant differences between female and male AKI cases (*p*-value n.s. ≥ 0.05, * <0.05).

**Table 5 jcm-12-01092-t005:** Comparison of the proportions of progressive AKI cases within different age groups, stratified by sex. A total of 7592 AKI cases were considered for detection of progression. *n*—number of cases. Significant *p*-values (< 0.05) highlighted as bold.

Criteria Feature	AKIN1, *n*, %			AKIN3, *n*, %		
	Female (*n* = 6011)	Male (*n* = 9365)	*p*-Value	Female (*n* = 1032)	Male (*n* = 1984)	*p*-Value
Absolute	1214, 20.2%	2750, 29.4%	**<0.001**	138, 13.4%	302, 15.2%	0.190
Both	1450, 24.1%	2510, 26.8%	**<0.001**	294, 28.5%	936, 47.2%	**<0.001**
Relative	3347, 55.7%	4105, 43.8%	**<0.001**	600, 58.1%	746, 37.6%	**<0.001**

## Data Availability

Data are available upon reasonable request to the corresponding author.

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
