# Peer review of "Laboratory Diagnostic of Acute Kidney Injury and Its Progression: Risk of Underdiagnosis in Female and Elderly Patients"

_jcm, 2023, doi:10.3390/jcm12031092_

Round 1

Reviewer 1 Report

Every empirical calculation such as CKD-EPI formula should be questioned and improved on regular basis. Authors addressed the problem of underdiagnosed AKI in female and elderly patients.

Using a large pool of data they proved the fact, which is already known to many clinicians. Although this proof is scientifically important, there is a room for improvement.

From the enviably large pool of data authors could give some suggestions how to improve or correct the present CKD-EPI formula, which is in use for more than a decade.

This would significantly add the value to this manuscript and increase its clinical importance.

Reviewer 2 Report

Reviewer’s opinion

It was very interesting because it was a study on what I usually felt as a clinician. In fact, in women, especially the elderly, AKI is often underestimated due to its low sCr.

[Major opinion]

1. However, this study raises a fundamental question. I think gender is a variable that cannot be corrected according to one gender, and I think that the difference between men and women should be recognized in the diagnostic process.

Since the emergence of the eGFR calculation method, the eGFR calculation method has been advanced through various versions in consideration of age and gender differences.

Does the re-defining of men or young men insist on returning to the initial eGFR calculation method?

If so, it is thought to be the result of ignoring diversity. There is a great concern that it can be understood that the change in the calculation method of eGFR so far is meaningless.

2. It is true that the sCr value in the elderly or women lacks trust due to muscle mass, and the disucussion also mentioned this, so shouldn't a new definition emerge based on muscle mass?

I think that deriving the AKI algorithm by measuring muscle mass will help overcome the direct association and the difference caused by muscle mass.

Or, in order to construct a new definition of AKI in elderly patients, I think it would be better to construct a new definition or algorithm of AKI related to muscle mass by measuring eGFR with inulin for each AKI stage.

3. Despite preaching the importance of muscle mass, there is no analysis of muscle mass in the study. There is not even a mention of Wt.

It is regrettable that there is no mention of weight change and muscle loss during the hospitalization period, even if it does not speak for muscle mass.

Patients with AKI have an average hospital stay of about 24 days, so it can be expected that there has been a sufficient weight change.

4. Is it right to define sex as bias?

I know that sex bias means discriminating between men and women in treatment or making a difference in the definition of terms, but I wonder if there are references for defining

The difference between men and women as bias in the diagnostic method.

5. Is it because the statistical method used the M-W method, but the parameters were sufficient, but it was not a normal distribution?

I think it is better to use the regression model by inserting various variables, not just comparing the occurrence of AKI.

It is regrettable that the K-M method or cox regression could have been used for mortality.

6. I think the impact of comobidity cannot be ignored, but it may be a method to analyze age, comobidity, etc. by matching.

7. The causes of AKI are very diverse, and there are surgery, drugs, and so on, and there are many things that affect motality such as blood pressure, weight, electrolyte, and acid base balance, but all of those variables were excluded.

Since it is a single center study, it is believed that the variables could have been used for analysis, and it is better to perform a regression analysis including this.

Round 2

Reviewer 2 Report

The authors gave answers on various references, but I still think this study is not appropriate as a clinician.

1) I hope that the authors will once again be aware of the causes of underdianosis of the occurrence of AKI in women and the elderly. As the cause, most of them relate to protein intake or muscle mass, so it is not a matter that can be expressed simply arithmetically. Also, this is not just a matter of sarcopenia, but the difference between sex and ageing. The elderly have less muscle mass than the young, but not all of them are sarcopenia. If AKI is under-diagnosed due to the problem of muscle mass, I think it is right to study how to correct the difference in musle mass between men and women to create standards. It is thought to be making a new attempt at changing the standard of muscle mass of women to men and muscle mass of elderly men to young men, but as a clinician, I cannot agree with this method.

2) As the authors say, there are many errors in judging AKI simply by sCr. That's why the eGFR formula came out and it's already a formula that considers age and gender. The articles cited as evidence acknowledged and analyzed gender differences. It is not an argument to use one gender as a criterion, since there is a gender difference.

As the authors argue, if you are asking to find a new criteria for the AKI diagnosis of women and the elderly because there is a limit to sCr, I repeated, I think it is right to find different criteria for each woman and the elderly, not to eliminate gender differences.

3) Measurement of weight is necessary. It may be difficult and inaccurate to measure, and it may not represent muscle mass, but if there is a wt gain such as edema, it is judged that weight information should be used in the form of exclusion through chart review.

This study is a new attempt at the definition of AKI in old age and women, and it is considered that sufficient experimental research and evidence are inevitably needed because it is inevitably controversial.

Trying to define another AKI without sufficient grounds with only EMR data can cause many problems.

Author Response

Response to the second Review Report

The authors gave answers on various references, but I still think this study is not appropriate as a clinician.

- Response: Our study cannot be a panacea for eGFR-based renal function diagnostics in clinical practice and is not meant to be. The fact is that AKI criteria are widely and often uncritically applied. On the basis of a very large cohort our study shows impressively how relevant the influencing factors sex and age of the patients are, sensitizes for these diagnostic limitations and offers a first approach for a more precise detection, which, considered in isolation, cannot cure the limitations of indirect eGFR-based diagnostics. For the above reasons, we consider the publication of our data to be very important. We have addressed the helpful comments in detail below, for which we would like to express our thanks.

1) I hope that the authors will once again be aware of the causes of underdiagnosis of the occurrence of AKI in women and the elderly. As the cause, most of them relate to protein intake or muscle mass, so it is not a matter that can be expressed simply arithmetically. Also, this is not just a matter of sarcopenia, but the difference between sex and ageing. The elderly have less muscle mass than the young, but not all of them are sarcopenia. If AKI is under-diagnosed due to the problem of muscle mass, I think it is right to study how to correct the difference in musle mass between men and women to create standards. It is thought to be making a new attempt at changing the standard of muscle mass of women to men and muscle mass of elderly men to young men, but as a clinician, I cannot agree with this method.

- Response: We thank this reviewer for this comment. The present study aimed to quantify a potential sex-related difference in the detection of AKI. Indeed, the study found such difference indicating more AKI in men compared to woman. We speculate that such difference may be related to muscle mass and protein intake. However, as this study did not aim to explore causes of the underdiagnosis of AKI in women compared to men, we suggest leaving this question open for further studies. We agree: As long as we rely on sCr for the diagnosis of CKD as well as AKI, a correction for muscle mass is necessary. A first correctional approach is the inclusion of sex and age in the equations, as these two well-documented variables explain a relevant part of interindividual muscle mass differences. For CKD this has already been implemented in the eGFR equations. However, it is not implemented in the AKI definition yet. This is why we see the implementation of age and sex in the AKI definition as one first step towards a better correction for muscle mass and therefore more accurate AKI diagnoses. The clinicians in our working-group feel that the results of the present study are substantial given newly reported quantification of potential AKI underdiagnosis in women compared to men, a new published result. Having this study in mind, clinicians may be more aware that the current serum creatinine-based AKI definition scoped by the KDIGO AKI criteria disadvantages women and the elderly, a fact not clearly demonstrated before the present study.

2) As the authors say, there are many errors in judging AKI simply by sCr. That's why the eGFR formula came out and it's already a formula that considers age and gender. The articles cited as evidence acknowledged and analyzed gender differences. It is not an argument to use one gender as a criterion, since there is a gender difference.

- Response: We agree with the reviewer comment in that we would not want to eliminate sex differences from AKI criteria. In fact, the opposite may be true. We recommend more studies, which modify AKI criteria for sex. In addition, the systematic and automated consideration of muscle mass e.g. from radiological diagnostics and other clinical factors in addition to the determination of appropriate diagnostic biomarkers offers potential for future research questions with regard to more precise diagnostics and clinical decision support systems, which we want to explore in future studies.
Also, we think there are two terms mixed up here. Yes, the eGFR equation was developed to give a rough estimate of kidney function and considers age as well as sex. Although it has a considerable margin of error, it seems applicable in many patient cases, especially in assessing long term developments of kidney function such as in CKD and its staging. We, on the other hand, are looking at the diagnostic process of short-term kidney function developments such as AKI, which is currently solely based on sCr values, without any correction for age or sex. This process was agreed upon by the International Society of Nephrology through the KDIGO Clinical Practice Guideline. An age or sex adjustment in the diagnostic process is only mentioned in one context throughout the whole guideline: An estimated baseline (not the actual baseline per patient) sCr can be deduced by calculating sCr from a set eGFR using various age, sex, and race categories. This however does not circumvent the problem of the overall potential sCr increase which should also be different according to the muscle mass, for which age, sex and race are proxies. The guideline even gives the following conclusion: Better delineation of the effects of age on the risk for AKI is needed.

As the authors argue, if you are asking to find a new criterion for the AKI diagnosis of women and the elderly because there is a limit to sCr, I repeated, I think it is right to find different criteria for each woman and the elderly, not to eliminate gender differences.

- Response:

Indeed, asking for the development of new criteria is the main point we are trying to make. We do not suggest elimination of sex differences and are somewhat confused as to where in the manuscript this point is specifically mentioned.

3) Measurement of weight is necessary. It may be difficult and inaccurate to measure, and it may not represent muscle mass, but if there is a wt. gain such as edema, it is judged that weight information should be used in the form of exclusion through chart review.

This study is a new attempt at the definition of AKI in old age and women, and it is considered that sufficient experimental research and evidence are inevitably needed because it is inevitably controversial.

Trying to define another AKI without sufficient grounds with only EMR data can cause many problems.

- Response:

We agree that a new AKI definition should not be based solely on EMR data. This is also not the intention of this paper. With the analysis of EMR data we have been able to show that the current AKI definition without consideration of sex and age bears the risk of underdiagnosis of patient groups with low muscle mass, such as women and the elderly. In the revised manuscript, we clearly state that the present study aimed to quantify a potential sex-related difference in the identification of AKI. Indeed, the study found such differences indicating more AKI in men compared to woman but did not explore causes of the underdiagnosis. We suggest considering weight together with other reasonable aspects mentioned by the reviewer for future research in this field.
Again, we have to stress the following point: we are not trying to create a new definition of AKI. We have included the following statements into the manuscript to clearly stress this fact and prevent any kind of misinterpretation: “These findings should raise awareness not to uncritically apply diagnostic criteria to all patients regardless of sex, age, as well as other criteria such as muscle mass in a clinical context. This could help to ensure that patients with an increased risk of kidney injury underestimation receive a more equitable diagnosis through additional laboratory pa-rameters (e.g., CysC) and to provide adequate therapy. We want to emphasize that our study does not propose a new AKI definition which is currently clinically applicable. Ra-ther, we wish to reiterate that the current criteria for AKI diagnosis may lead to possible underdiagnosis. We believe that the criteria for AKI diagnosis should be expanded to in-clude a factor that compensates for sex or that a measure of filtration that is unlikely to be influenced by sex should be used. Further studies in this regard are urgently needed.” (page 9, line 290-300)